# Increased Plasma Levels of Gut-Derived Phenolics Linked to Walking and Running Following Two Weeks of Flavonoid Supplementation

**DOI:** 10.3390/nu10111718

**Published:** 2018-11-09

**Authors:** David C. Nieman, Colin D. Kay, Atul S. Rathore, Mary H. Grace, Renee C. Strauch, Ella H. Stephan, Camila A. Sakaguchi, Mary Ann Lila

**Affiliations:** 1Human Performance Laboratory, Appalachian State University, North Carolina Research Campus, Kannapolis, NC 28081, USA; 2Food Bioprocessing and Nutrition Sciences, Plants for Human Health Institute, North Carolina State University, North Carolina Research Campus, Kannapolis, NC 28081, USA; cdkay@ncsu.edu (C.D.K.); asrathor@ncsu.edu (A.S.R.); mhgrace@ncsu.edu (M.H.G.); rcstrauc@ncsu.edu (R.C.S.); mlila@ncsu.edu (M.A.L.); 3Department of Nutrition, UNC Gillings School of Global Public Health, University of North Carolina-Chapel Hill, NC 27599, USA; estephan@email.unc.edu; 4Physical Therapy Department, Federal University of São Carlos, São Carlos, SP 13565-905, Brazil; camilasakaguchi790@gmail.com

**Keywords:** exercise, polyphenol, metabolite, hippurate, intestinal tract, colon

## Abstract

Using a randomized, double-blinded, placebo-controlled, parallel group design, this investigation determined if the combination of two weeks of flavonoid supplementation (329 mg/day, quercetin, anthocyanins, flavan-3-ols mixture) and a 45-minute walking bout (62.2 ± 0.9% VO_2max_ (maximal oxygen consumption rate)) enhanced the translocation of gut-derived phenolics into circulation in a group of walkers (*n* = 77). The walkers (flavonoid, placebo groups) were randomized to either sit or walk briskly on treadmills for 45 min (thus, four groups: placebo–sit, placebo–walk, flavonoid–sit, flavonoid–walk). A comparator group of runners (*n* = 19) ingested a double flavonoid dose for two weeks (658 mg/day) and ran for 2.5 h (69.2 ± 1.2% VO_2max_). Four blood samples were collected (pre- and post-supplementation, immediately post- and 24 h post-exercise/rest). Of the 76 metabolites detected in this targeted analysis, 15 increased after the 2.5 h run, and when grouped were also elevated post-exercise (versus placebo–sit) for the placebo– and flavonoid–walking groups (*p* < 0.05). A secondary analysis showed that pre-study plasma concentrations of gut-derived phenolics in the runners were 40% higher compared to walkers (*p* = 0.031). These data indicate that acute exercise bouts (brisk walking, intensive running) are linked to an increased translocation of gut-derived phenolics into circulation, an effect that is amplified when combined with a two-week period of increased flavonoid intake or chronic training as a runner.

## 1. Introduction

Small amounts of flavonoids from ingested foods and beverages are absorbed in the small intestine [1]. A much larger proportion of ingested flavonoids, however, remain unabsorbed and persist in the lower intestine for long periods of time, where they experience microbial degradation (including ring fission) and form a diversity of phenolic compounds that can be absorbed, undergo phase II metabolism, and exert a variety of bioactive effects before elimination in the urine. 

In a previous 17-day study using a supplement with blueberry and green tea flavonoids, vigorous exercise significantly increased the translocation of gut-derived phenolics into the blood compartment [2]. This study, however, used an untargeted metabolomics approach that was not optimized to detect the diversity of gut-derived flavonoid metabolites as reported in recent literature.

Exercise has been linked in many studies to a transient increase in gut permeability, an effect that can occur after just 20 min of vigorous exercise [3,4]. In exercise-based studies, whole gut permeability is commonly measured using the ratio of lactulose and mannitol sugars in the urine. A transient increase in gut permeability has been speculated as a possible mechanism by which vigorous exercise accelerates the movement of beneficial gut-derived phenolics from the lower intestine into the blood [5]. An alternative hypothesis is that moderate- and high-intensity exercise may influence the activity of transporters that control the movement of flavonoids and their transformed metabolites across the wall of the gastrointestinal tract into circulation. In one study, urinary excretion of microbial phenolic metabolites was higher during exercise training compared to during a week of no training in 10 endurance-trained males [6]. Taken together, the limited data available suggest that plasma levels of gut-derived phenolics may be linked to both chronic and acute exercise influences.

Using a randomized, parallel group design, this investigation determined if the combination of two weeks of flavonoid supplementation (329 mg/day) and one acute 45-minute brisk walking bout (70% VO_2max_) enhanced the translocation of gut-derived phenolics into the circulation in a group of healthy walkers (*n* = 77). A comparator group of trained runners (*n* = 19) ingesting a double dose of the flavonoid supplement for two weeks was included (2.5 h run, 70% VO_2max_). A secondary analysis compared pre-study plasma levels of gut-derived phenolics in the groups of walkers and runners. A key feature of this study was the use of a targeted metabolite analysis protocol that had been optimized and validated to detect and quantify 121 analytes relative to 75 commercial and synthetic reference standards. 

## 2. Materials and Methods 

### 2.1. Participants 

Healthy males and females with a history of regular walking (>100 min per week), with a body mass index (BMI) less than 35 kg/m^2^, and of 18 to 50 years of age (*n* = 81) were entered into the study and randomized to the flavonoid or placebo supplement groups for two weeks. A total of 77 of these participants completed all phases of the study: placebo–sit (*n* = 16), placebo–walk (*n* = 20), flavonoid–sit (*n* = 20), and flavonoid–walk (*n* = 21). Male and female runners (*n* = 21) with a history of participating in 10 km to 42.2 km races and the ability to run for 2.5 h on a treadmill in a laboratory setting were entered into the study as a comparator group (i.e., to compare both acute and chronic exercise responses on gut-derived phenolics between groups). A total of *n* = 19 runners completed all phases of the study. The four walkers and two runners that did not complete all study requirements dropped out due to changes in schedules. All study participants agreed to consume less than five servings/day of fruits and vegetables, less than two cups/day coffee, and no green tea during the two-week study. Study participants also agreed to avoid use of nonsteroidal anti-inflammatory drugs (NSAIDs) and all dietary and herbal supplements during the two-week study. Participants signed the informed consent, and procedures were approved by the university Institutional Review Board. (Trial registration: ClinicalTrials.gov, U.S. National Institutes of Health, identifier: NCT03249571.)

### 2.2. Research Design

This study utilized a randomized, parallel group design (Figure 1).

Study participants reported to the research facility for baseline testing and orientation, then pre-study and after two weeks of supplementation, and then again 24 h later. Four blood samples were collected (pre- and post-supplementation, post-exercise/rest, and 24-h post-exercise/rest). When reporting to the lab after two weeks of supplementation, study participants were randomized to either sit or walk briskly on treadmills for 45 min. A comparator group of 20 runners ingested a double dose of the flavonoid supplement for two weeks, and then ran on treadmills for 2.5 h in the laboratory. 

#### 2.2.1. Supplements

Supplements for the walking group were administered in a double-blinded manner in capsule form. Supplement and placebo capsules were prepared by Reoxcyn LLC (Pleasant Grove, UT, USA). Supplement ingredients (US Patent 9,839,624) included the following (in 2 capsules) and provided 329 mg total monomeric flavonoids: 100 mg vitamin C (as ascorbyl palmitate) (Green Wave Ingredients, La Mirada, CA, USA), wild bilberry fruit extract with 64 mg anthocyanins (FutureCeuticals, Momence, IL, USA), green tea leaf extract with 184 mg total flavan-3-ols (Watson Industries, Inc., Pomona, CA, USA), 104 mg quercetin aglycone (Novel Ingredients, East Hanover, NJ, USA), 107 mg caffeine (Creative Compounds, Scott City, MO, USA), and 60 mg omega 3 fatty acids (Novotech Nutraceuticals, Ventura, CA, USA). Capsule fill ingredients and excipients included Nu-Flow 70R (from rice hulls), tapioca from cassava root, natural bamboo silica, and marshmallow root. Placebo capsules contained only the fill ingredients and excipients (without the active ingredients). 

As previously reported, the capsule contents were analyzed prior to the study for flavonoid content using high-performance liquid chromatography (HPLC) [7]. The flavonoid content was calculated as the sum of anthocyanins (measured as cyanidin-3-*O*-glucoside equivalents), quercetin, and flavan-3-ol compounds (epicgallocatechin gallate (EGCG), epicatechin, epigallocatechin, and epicatechin gallate).

The daily serving for the walkers was 2 flavonoid (329 mg/day) or 2 placebo capsules. The daily serving for the runners was 4 flavonoid capsules (658 mg/day). Participants were given a two-week supply of either the flavonoid or placebo capsules, with instructions on how to consume the capsules daily in split doses (1 with breakfast and 1 with lunch, or double that amount for the runners). Study participants ingested 2 capsules (4 for the runners) just after providing the blood sample during the walk/rest (or run) lab session. 

#### 2.2.2. Orientation and Pre-Study Measurements 

One to two weeks before the start of the study, study participants (walkers and runners) were given an orientation to the study. Instructions were given for recording all food and beverage intake in 3-day food logs (Thursday, Friday, and Saturday prior to starting supplementation). Demographic and training histories were acquired with questionnaires. Height, body weight, and percent body fat were measured (seca Medical Body Composition Analyzer 514 bioelectrical impedance scale, Hanover, MD, USA). VO_2max_ was assessed using the Bruce’s treadmill protocol, with oxygen consumption and ventilation continuously monitored using the Cosmed CPET metabolic system (Cosmed, Rome, Italy). 

On the first day of the two-week supplementation period, participants (walkers and runners) returned to the lab in a fasted state (nine or more hours with no food or beverage other than water). Blood samples were taken from an antecubital vein with subjects in the seated position. Participants (walkers) were given a two-week supply of flavonoid or placebo capsules organized into supplement trays to facilitate compliance. Runners were given a two-week supply of flavonoid capsules in supplement trays. The 3-day food record was turned in and reviewed by the research team, and analyzed for nutrient and flavonoid content using the Food Processor Version 11.1 (ESHA Research, Salem, OR, USA). ESHA’s port utility (Version 4.0; ESHA Research, Salem, OR, USA) was used to upload the Flavonoid Values for USDA Survey Foods and Beverages (FNDDS) 2007–2010 database [8]. Each food/beverage was assessed for macro- and micro-nutrients, total flavonoids and subtotals for each of the six flavonoid subclasses, and three individual flavonoid values (quercetin, cyanidin, and epigallocatechin gallate (EGCG)).

#### 2.2.3. Laboratory Exercise Sessions

After the two-week supplementation period, participants (walkers and runners) returned to the lab in an overnight fasted state (on the same day of the week as the pre-supplementation lab visit). Participants turned in the supplement tray to verify compliance to the supplementation regimen. A blood sample was collected. Participants (walkers) were then randomized to either sit for 45 min in the lab or to walk briskly for 45 min on a 5% graded treadmill at 60% VO_2max_ (with metabolic monitoring during the first 5 min, and then at 15, 30, and 45 min) using the Cosmed CPET metabolic cart (Cosmed, Rome, Italy). Runners ran at 70% VO_2max_ on an ungraded treadmill for 2.5 h, with metabolic measurements made every 30 min. Water was given ad libitum for all participants, with no other beverage or food allowed. 

Just prior to the sitting or walking lab sessions, and the 2.5 h run, participants consumed a 150 mL solution containing 1 g lactulose and 0.5 g mannitol (as markers of gastric permeability). An increase in the lactulose/mannitol ratio (L/M) was used as an indicator of increased gut permeability. The normal rate of absorption is approximately 10% for mannitol and less than 1% for lactulose. When gut permeability rises, lactulose absorption increases disproportionately to mannitol (i.e., an increase in L/M) [9,10]. Urine was collected in a plastic container for five hours after ingesting the sugar mixture. No eating or drinking (except tap water) was allowed during the 5-h period of urine collection. Participants returned the next morning in an overnight fasted state, provided a blood sample, and turned in the urine sample. 

### 2.3. Analytical Methods

#### 2.3.1. Targeted Metabolomics Analysis

##### Materials

Seventy-five commercially available and synthetic reference standards were purchased from Alfa Aesar (Tewksbury, MA, USA), Ark Pharm (Libertyville, IL, USA), Biovision (San Francisco, CA, USA), Chromadex (Irvine, CA, USA), Extrasynthese SA (Z.I Lyon Nord, France), Fisher Scientific (Waltham, MA, USA), Matrix Scientific (Columbia, SC, USA), Oxchem (Wood Dale, IL, USA), PhytoLab GmbH & Co. KG (Vestenbergsgreuth, Germany), Polyphenols AS (Sandnes, Norway), Sigma (St. Louis, MO, USA), TCI America (Portland, Oregon, USA), and Toronto Research Chemicals (Toronto, Ontario, Canada), and synthesized in a project sponsored by the Biotechnology and Biological Sciences Research Council (BBSRC) (BB/I0066028/1). (See Appendix A.)

##### Ultra Performance Liquid Chromatography–Tandem Mass Spectrometer (UPLC–MS/MS) Methodology

Gut-derived phenolic metabolites were purified from 100 μL plasma by 96-well solid-phase extraction (SPE; Strata™-X Polymeric Reversed Phase, Phenomenex, Torrance, CA, USA; microelution 2 mg/well). Extracts were separated and quantified via liquid chromatography tandem MS/MS. Briefly, HPLC–electrospray ionization (ESI)–MS/MS analysis was performed using a SCIEX QTRAP (SCIEX, Framingham, MA, USA) 6500^+^ enhanced high-performance hybrid triple quadrupole–linear ion trap mass spectrometer with an electrospray IonDrive Turbo V Source (SCIEX, Framingham, MA, USA) coupled to an Exion high-performance UHPLC (Exion high-performance UHPLC), with samples injected onto a Kinetex PFP UPLC column (1.7 µm particle size, 100Å pore size, 100 mm length, 2.1 mm internal diameter; Phenomenex, Torrance, CA, USA) with oven temperature maintained at 37 °C. Mobile phase pump A comprised 0.1% *v*/*v*. formic acid in water (Optima grade, Fisher Scientific) and pump B 0.1% *v*/*v* formic acid in LC–MS grade acetonitrile (Honeywell Burdick and Jackson, Muskegon, MI, USA), with binary gradient from 2% B to 90% B over 30 min and flow rate gradient ranging from 0.55 mL/min to 0.75 mL/min. 

MS/MS scanning was achieved via advanced scheduled multiple-reaction monitoring (ADsMRM) using positive and negative ionization mode toggling in Analyst (Version 1.6.3, SCIEX, Framingham, MA, USA) with quantitation conducted using MultiQuant (Version 3.0.2, SCIEX, Framingham, MA, USA) software platforms. Internal standards included L-tyrosine-^13^C_9_,^15^N, resveratrol-(4-hydroxyphenyl-^13^C_6_), and phloridzin dehydrate (Sigma-Aldrich), and 12-point calibration curves (1 ηM to 100 μM) were established using reference standards in a matched matrix (SPE extracted Corning pooled healthy donor Human AB Serum (# 35-060-CL, Mediatech Incorporated, Manassas, VA, USA). Source parameters included curtain gas 35, ion-spray voltage 4000, temperature 550, nebulizer gas 70, heater gas 70, and with optimized analyte specific quadrupole voltages (mean ± SD) in the range 40 ± 24 for declustering potential (min 4.5 to max 185), 10 ± 1 for entrance potential (min 3 to max 13), 26 ± 11 for collision energy (min 5 to max 59), and 14 ± 10 for collision cell exit potential (min 1 to max 51).

##### MS/MS Optimization

The targeted metabolite analysis protocol was optimized and validated to detect 121 analytes, which were quantified relative to 75 authentic commercial and synthetic standards. Reference standards were optimized for UPLC–MS/MS parameters (CV ≤ 15) with extraction efficiencies between 80% and 100% recoveries. Where reference standards for metabolites (including Phase II conjugates) were not available (46 analytes), identification was based on fragmentation profiling involving the precursor structure and 3–5 product transitions, and confirmed in pooled samples. These metabolites were then quantified relative to their closest structural reference standard with similar ionization intensities (Appendix A). Finally, all metabolites were confirmed on the basis of established retention times (using authentic and synthesized standards where possible) and three or more precursor-to-product ion transitions.

#### 2.3.2. Urine Sugar Analysis

Reference standards for lactulose and mannitol were purchased from Sigma-Aldrich and an internal standard, (UL-13C6glc)-Sucrose, was purchased from Omicron Biochemicals, Inc (South Bend, IN, USA). All standards were optimized and validated for UPLC–MS/MS parameters, and sugars in urine were quantified relative to authentic commercial standards. Samples from each treatment group and time point were pooled and a 1:100 dilution was prepared with 90% acetonitrile. Samples were separated and quantified via UPLC–MS/MS. Briefly, UPLC–ESI–MS/MS analysis was performed using a Waters Xevo G2-XS QTOF mass spectrometer (Waters Corporation, Milford, MA, USA) with a LockSpray source (Waters Corporation, Milford, MA, USA) and an ESI probe coupled to an ACQUITY I-Class UPLC (Waters Corporation, Milford, MA, USA). 

Samples were injected onto a Luna NH2 column (3 µm particle size, 100 Å pore size, 100 mm length, 2 mm internal diameter; Phenomenex) with column temperature maintained at 30 °C. Mobile phase A comprised 0.1% *v*/*v* formic acid in water (LC–MS grade), while mobile phase B comprised 0.1% *v*/*v*. formic acid in acetonitrile (LC–MS grade) (Honeywell International Inc., Morris Plains, NJ, USA. The binary gradient ranged from 95% B to 20% B over 7.5 min with a flow rate of 0.7 mL/min. MS/MS scanning was achieved via time of flight multiple reaction monitoring (Tof-MRM) (Waters Corporation, Milford, MA, USA) with target enhancement using the negative ionization mode in MassLynx (Version 4.1, Waters Corporation, Milford, MA, USA) with quantitation performed using TargetLynx (Version 4.1, Waters Corporation, Milford, MA, USA) software platforms. Ten-point calibration curves (from 1 nM to 5 μM for lactulose and from 39 nM to 150 μM for mannitol) were established using reference standards in a matched (negative control) matrix (SurineTM Negative Urine Control, Certified Reference Material, Sigma Aldrich). Source parameters included capillary voltage 1.00 kV, sampling cone 50–70 (arbitrary units, analyte dependent), source offset 80 (arbitrary units), source temperature 150 °C, desolvation gas temperature 600 °C, cone gas flow 50 L/h, desolvation gas flow 1200 L/h, and collision energy 6–10 (analyte dependent). The lactulose/mannitol ratio was used as an indicator for small and large intestine permeability.

### 2.4. Statistical Procedures

Data are presented as mean ± standard error (SE). Comparisons between the walker groups and runners (subject characteristics, performance data, nutrient intake, and other selected variables) were compared using one-way analysis of variance (ANOVA) with post-hoc tests using Tukey’s HSD, and statistical differences were accepted when the *p*-value was ≤0.05. The plasma metabolite data (both single metabolites and grouped metabolites) were analyzed using the generalized linear model (GLM) and a 5 (groups) × 4 (time) repeated-measures ANOVA with a between-participants design (IBM SPSS Statistics for Windows, Version 24.0, IBM Corp, Armonk, NY, USA). When the interaction statistic was significant in the GLM analysis (*p* ≤ 0.05), post-hoc analyses were conducted with Student’s *t*-tests comparing the change from pre-study values over time against the placebo–sit group. For these analyses, statistical differences were accepted when the *p*-value was ≤0.017. 

## 3. Results

The analysis included 77 walkers divided into four subgroups (placebo–sit (*n* = 16), placebo–walk (*n* = 20), flavonoid–sit (*n* = 20), flavonoid–walk (*n* = 21)), and 19 runners who successfully adhered to all aspects of the study design (Table 1). No significant differences were found for any of the variables listed in Table 1 among the five groups except for significantly higher maximal oxygen consumption rates (VO_2max_) and lower body weight and body fat percentages for the runner group compared to each of the walker groups. 

Three-day food records collected before the study began revealed no significant group differences using one-way ANOVA in terms of energy, macronutrient, micronutrient, and total flavonoid intake except for cholesterol (*F* = 3.05, *p* = 0.021, with Tukey’s HSD post-hoc tests showing no significant group differences) (Appendix A). Total flavonoid intake was relatively low [8] for both the walkers and runners (105 ± 12.0 and 56.1 ± 15.0 mg/day, respectively, *p* = 0.08), with no group differences for the six flavonoid subgroups (anthocyanins, flavan-3-ols, flavonols, flavanones, isoflavones, flavones) or specific flavonoids (cyanidin, EGCG, quercetin). The two-week flavonoid supplementation regimen did not significantly increase fasting plasma gut-phenolic levels (average of all 76 metabolites detected) compared to placebo (2.19 ± 1.48 µM and −0.45 ± 1.24 µM, respectively, *p* = 0.185) in the study participants from the walking groups. Overnight fasted plasma hippuric acid levels did increase significantly in the runners (658 mg flavonoids/day) versus the placebo–sit group (329 mg flavonoids/day) as depicted in Figure 2 (*p* = 0.004).

Table 2 summarizes the performance data for the 41 walkers randomized to the 45-minute walking bout and the 19 runners who ran for 2.5 h on treadmills. Absolute and relative oxygen consumption and heart rates were significantly higher, as designed, during the running compared to the walking bouts.

Figure 3 summarizes the lactulose/mannitol ratios (L/M) measured from pooled urine samples (collected for 5 h after sugar ingestion) for each of the five groups. L/M was 33% higher in the runner group compared to the placebo–sit group, and 19%, 45%, and 35% lower in the placebo–walk, flavonoid–sit, and flavonoid–walk groups, respectively. 

Post-exercise elevations (from pre-supplementation levels) in metabolites detected (Appendix A) were evaluated in the double-flavonoid-dose runner group relative to the placebo–sit group and selected for secondary analysis where trends were observed (group contrasts, *p* < 0.175). Fifteen metabolites were identified, grouped (to improve the ability to detect intervention effects), and compared between the four study groups and runner comparator group. Figure 4 shows that higher post-exercise changes (from pre-supplementation levels) were observed (compared to the placebo–sit group) for the placebo–walk, flavonoid–walk, and flavonoid–run groups. The 15 gut-derived phenolics were hippuric acid, 3-hydroxyhippuric acid, 4-hydroxycinnamic acid, 5-(3′,4′-dihydroxyphenyl)-γ-valerolactone, 4-hydroxy-3-methoxybenzoic acid, 4-hydroxybenzaldehyde, 3-methoxybenzoic acid-4-*O*-glucuronide, 4-methoxybenzoic acid-3-*O*-glucuronide, 3-(3-hydroxy-4-methoxyphenyl)propanoic acid-3-*O*-glucuronide, quercetin-3-*O*-glucuronide and delphinidin-3-*O*-glucoside, dihydroxybenzaldehyde-*O*-glucuronide, hydroxy-methoxybenzyldehyde-*O*-glucuronide, hydroxy-methoxybenzaldehyde sulfate, and trihydroxy-benzaldehyde sulfate. See Appendix A for more information on these metabolites. 

Figure 5 depicts group comparisons across the four time points for one of the 15 gut-derived phenolics (as an example) that increased after both walking and running relative to the placebo–sit group: 3-(3-hydroxy-4-methoxyphenyl) propanoic acid-3-*O*-glucuronide (interaction effect, *p* < 0.001). 

A secondary analysis evaluated the effect of fitness status on the pre-study plasma levels of the 76 identified metabolites in runners (*n* = 19) compared to the walkers (*n* = 77) (Figure 6). The average plasma level of these metabolites in the runners was 40% higher than in the walkers (*p* < 0.001, Figure 6A). An additional analysis showed that this difference was primarily driven by greater runner levels for seven plasma gut-derived phenolics (*p* < 0.001): 4-hydroxyphenylacetic acid, 5-*O*-caffeoylquinic acid, 5-*O*-feruloylquinic acid, 3,4-dihydroxycinnamic acid-4-*O*-glucuronide, pyridoxic acid sulfate, hydroxy-methoxyphenylacetic acid-*O*-glucuronide, and dihydroxy-benzaldehyde-*O*-glucuronide. Minor but significantly lower plasma levels of 32 metabolites were found in the runners (represented as orange stacked column bars; Figure 6B). 

## 4. Discussion

The data from this randomized, double-blinded, placebo-controlled, parallel group study with walkers (*n* = 77) and a comparator group of runners (*n* = 19) showed that the combination of two weeks of flavonoid supplementation and exercise (both 45 min brisk walking and 2.5 h running) enhanced the translocation of gut-derived phenolics into the circulation. Of the 76 gut-derived phenolic metabolites detected, 15 were found to be most responsive to acute exercise, with higher post-exercise changes measured (versus placebo–sit) for the placebo–walk, flavonoid–walk, and flavonoid–run groups. The pre-study plasma concentration of the gut-derived phenolic metabolites (all 76 that were detected) was 40% higher in the runners than in the walkers. These data indicate that acute exercise bouts (both brisk walking and intensive running) combined with flavonoid supplementation, and the elevated fitness status associated with habitual running, are linked to elevations in plasma levels of gut-derived phenolics. There was no discernable increase in overnight fasted plasma levels of gut-derived phenolics after the two-week supplementation period, due in part to the >17-h time period from the previous day’s flavonoid dose (morning, noon). The post-exercise increase in plasma levels of gut-derived phenolics appears to be related to a true exercise effect, as supported by the post-exercise increase observed in the placebo–walk group. However, the larger post-exercise increase in plasma levels of gut-derived phenolics measured in the flavonoid–walk and flavonoid–run groups relative to the other groups could also be related to the acute flavonoid dose ingested just prior to the exercise sessions. 

In a previous study, we showed that a 3-day period of intensive exercise (2.5 h running/day) enhanced the plasma gut-derived phenolic signature following a 17-day period of high flavonoid intake (blueberry and green tea extracts) [2]. Increases in plasma gut-derived phenolic metabolites linked to bacterial metabolism included increases in hippurate (1.8-fold) and 4-methylcatechol sulfate (2.5-fold), and elevations persisted for at least 14 h post-exercise. Increases were also measured, albeit to a lesser extent, for other similar metabolites including 2-hydroxyhippurate, 3-hydroxyhippurate, 4-hydroxyhippurate, catechol sulfate, and O-methylcatechol-sulfate. The primary limitation of this study was the use of an untargeted metabolomics approach with median-scaled intensity values. Also, the exercise regimen was unusually rigorous, with little transference value to the general population. In the current study, we sought to extend these findings by including two types of exercise modes and durations (45 min walking and 2.5 h running), a moderate dose of a diverse mixture of flavonoids (329 mg/day, two-week period), and the utilization of a quantitative targeted gut-derived phenolic metabolite analysis protocol optimized and validated using reference standards. 

The novel finding that the combination of increased flavonoid intake and moderate/vigorous exercise was associated with an acute, transient increase in plasma gut-derived phenolic metabolites could be due to several underlying mechanisms. These include increased gut permeability, a selective change in gut transporter density and function, changes in gut microbiota population diversity, and altered gastrointestinal motility and transport rate [6,11,12]. The relative importance of these various factors in explaining study results remains to be determined in future investigations. The urine sugar data (L/M) supported an increase in gut permeability following the 2.5 h running bout, but not the walking bouts. To the contrary, the L/M ratio was substantially lower for the two flavonoid groups (flavonoid–sit and flavonoid–walk) compared to the placebo–sit group. Flavonoids and their bio-transformed metabolites exert direct effects within the gastrointestinal tract, including maintenance of intestinal barrier integrity [13,14]. Flavonoids play a role in protecting intestinal epithelial cells from inflammation-induced permeabilization [15]. Prolonged and intensive running but not moderate walking increases IL-6 and other cytokines that disrupt the intestinal tight junction barrier and increase permeability [16,17]. Taken together, our data suggest that despite a two-week period of increased flavonoid intake, intensive and prolonged running was coupled with a post-exercise surge in plasma gut-derived phenolic metabolites due in part to increased gut permeability. Although speculative, the transient post-exercise elevation in circulating gut-derived phenolics may play a role in diminishing inflammation and oxidative stress during recovery from intensive running [18]. 

The modest increase in plasma gut-derived phenolic metabolites following two weeks of flavonoid supplementation and the 45-minute brisk walking bout may have occurred through other mechanisms including modifications in gut transporter localization, density, and function. Flavonoid absorption and distribution throughout the body are dependent on specific cell transport systems [1]. The solute carrier 22 (SLC22) transporter family regulates multiple metabolic pathways and signaling molecules including those related to gut microbiome products, tricarboxylic acid cycle intermediates, dietary flavonoids, nutrients, prostaglandins, and short-chain fatty acids [19]. The combined influence of flavonoid supplementation and acute exercise on SLC22 and other notable transporters in the Organic Anion Transporter (OAT) and ATP-binding cassette (ABC) (transporter families), however, is currently unknown. Limited rodent evidence indicates that exercise training upregulates the expression of ABC transporters [20]. 

The intestinal microbiota has an important role in the metabolism of flavonoids, and lifestyle interventions such as exercise and increased flavonoid intake have relatively rapid influences on microbial diversity [21,22,23]. Physical exercise and fitness promote an anti-inflammatory state, and this may be one of several mechanisms that enhance intestinal microbial diversity [24]. The gut microbiota is an essential component of flavonoid metabolism and diversifies when flavonoid ingestion increases [21]. Although gut microbial diversity was not measured in the current study, the greater concentration of plasma gut-derived phenolic metabolites in the runners and following acute exercise bouts in flavonoid-supplemented study participants may in part be coupled to greater intestinal microbial diversity. 

The pre-study cross-sectional comparison of the walkers and the leaner, more fit runners showed that plasma gut-derived phenolic levels were 40% higher in the runners. These are novel results that have not been previously reported. In another study, urinary excretion of colon-derived phenolic catabolites after orange juice intake (one acute 0.5 L dose) was slightly higher in the detrained compared to trained state in a small group of endurance-trained males [6]. Additionally, the urinary excretion rate for the trained males was lower than observed in a previous study with untrained volunteers. Our data are in opposition to these findings for several potential reasons including the use of different matrices (urine versus plasma) and, more importantly, study designs. Additionally, the current study measured a complex array of gut-derived phenolics versus a smaller panel focused on orange-juice-related metabolites. 

Flavonoid supplementation and the related transient surge in gut-derived phenolics from either walking or intensive running may over time lead to multiple health benefits (over and beyond the direct effects of exercise alone). There is increasing evidence that gut-derived phenolics have wide-ranging bioactive effects on multiple enzyme systems, exerting anti-inflammatory, anti-viral, and immune cell signaling influences, with enhancement of endothelial health and function in the intestine and vasculature [1,25,26,27,28,29,30]. Our data indicate that gut-derived phenolics circulate at higher levels throughout the body following flavonoid supplementation and exercise, potentially improving long-term health and reducing the risk for chronic diseases. 

To summarize from a public health viewpoint, this study utilized diet-relevant doses of flavonoids combined with acute walking bouts. These changes are achievable by a broad spectrum of the general population. This lifestyle stratagem was sufficient to transiently increase plasma levels of beneficial gut-derived phenolics, an effect that was heightened with acute and chronic running. 

## Figures and Tables

**Figure 1 nutrients-10-01718-f001:**
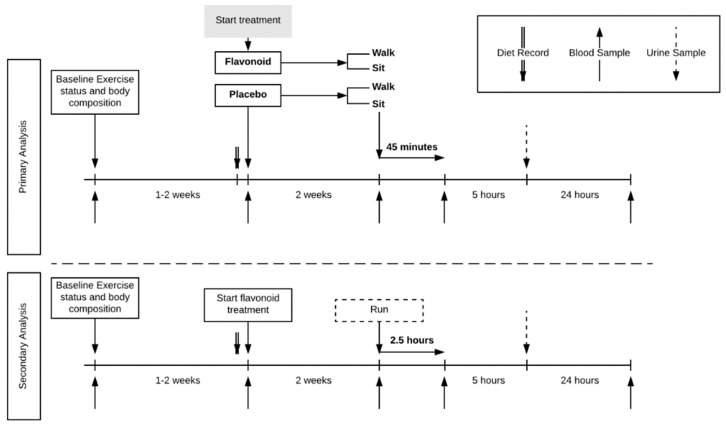
Trial design. Four-group parallel design with healthy habitual walkers (*n* = 77) randomized to treatment (flavonoid or placebo) and exercise (walk or sit). A comparator group of trained runners (*n* = 19) was included.

**Figure 2 nutrients-10-01718-f002:**
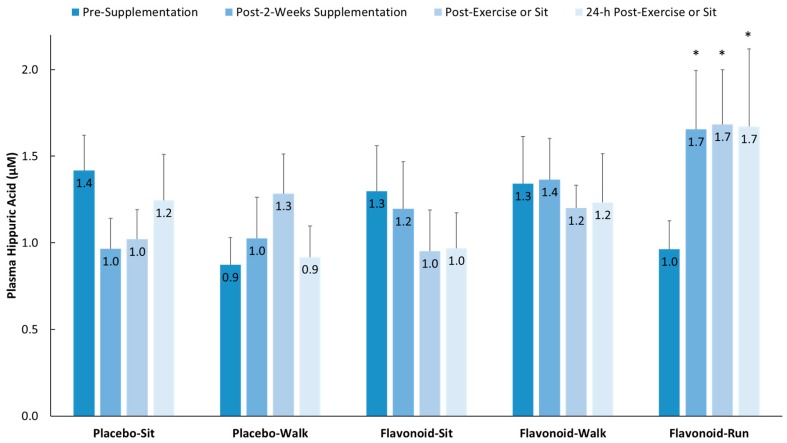
Plasma hippuric acid levels increased significantly in the flavonoid–run group compared to in the placebo–sit group. Interaction effect, *p* = 0.023; * *p* < 0.017, change from pre-supplementation versus placebo–sit. Data are means, with standard error represented as vertical lines.

**Figure 3 nutrients-10-01718-f003:**
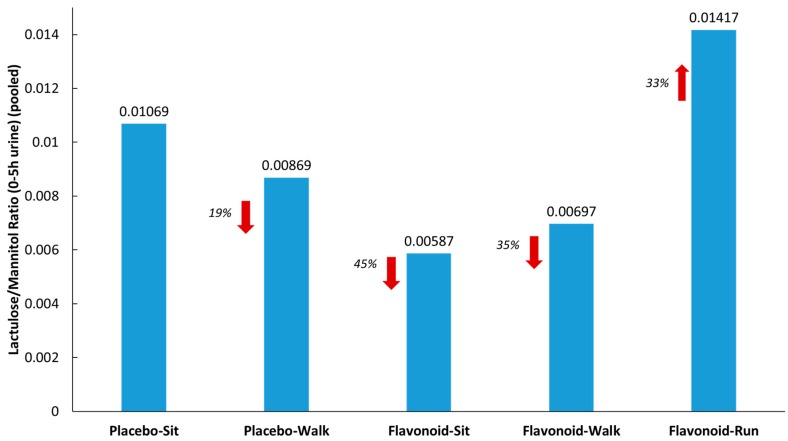
Lactulose/mannitol ratios from pooled urine (5 h collection) for each of the five groups. Red arrows indicate the % difference from the placebo-sit group.

**Figure 4 nutrients-10-01718-f004:**
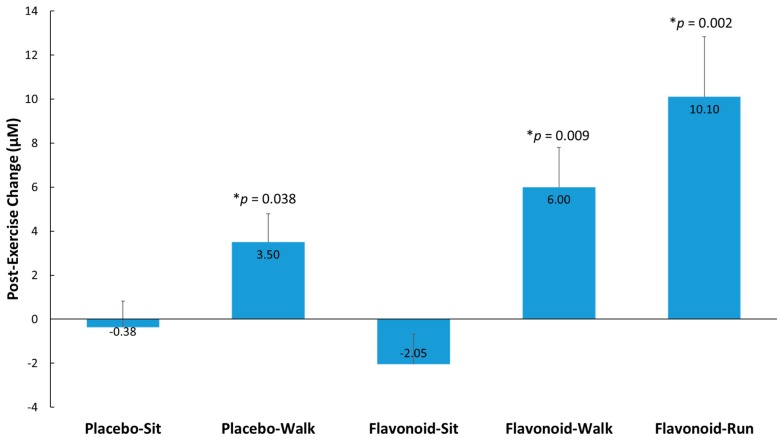
Post-exercise change from pre-supplementation for 15 selected and grouped plasma gut-derived phenolics (μM) with *p*-values indicated relative to the placebo–sit group. Interaction effect, *p* < 0.001. *p*-values represent significance testing for the contrast in change (post-exercise from pre-supplementation) compared to the placebo–sit group. Data are means, with standard error represented as vertical lines.

**Figure 5 nutrients-10-01718-f005:**
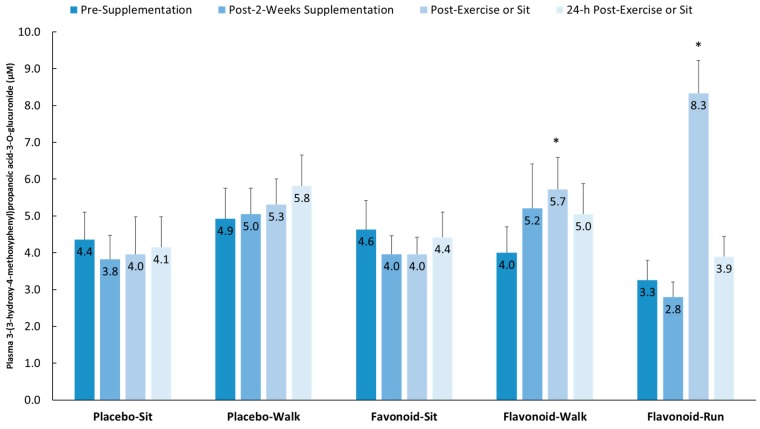
Plasma 3-(3-hydroxy-4-methoxyphenyl) propanoic acid-3-*O*-glucuronide increased significantly post-exercise (above pre-supplementation levels compared to the placebo–sit group) for both the flavonoid–walk and flavonoid–run groups. Interaction effect, *p* < 0.001; * *p* < 0.017. Data are means, with standard error represented as vertical lines.

**Figure 6 nutrients-10-01718-f006:**
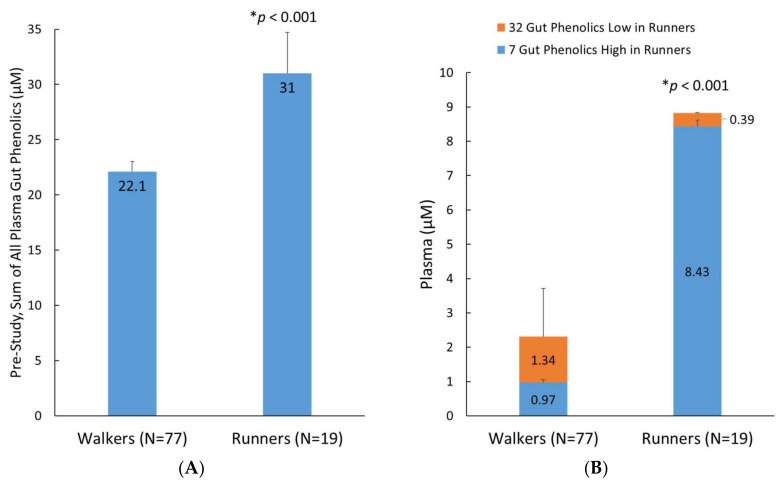
(**A**) Pre-study sum of all 76 plasma gut-derived phenolics detected in the analysis; (**B**) Pre-study group differences for 7 and 32 selected metabolites that were higher and lower, respectively, in the runners compared to the walkers. Data are means, with standard error represented as vertical lines.

**Table 1 nutrients-10-01718-t001:** Subject characteristics for each of the five groups (mean ± SE).

Subject Variables	Placebo–Sit (*n* = 16)	Placebo–Walkers (*n* = 20)	Flavonoid–Walkers (*n* = 21)	Flavonoid–Sit (*n* = 20)	Runners (*n* = 19)
Age (y)	38.7 ± 2.8	37.3 ± 2.6	36.3 ± 1.8	36.1 ± 2	36.7 ± 1.5
Height (cm)	168 ± 2.0	167 ± 2.0	170 ± 1.8	167 ± 2.0	172 ± 2.1
Weight (kg)	72.1 ± 4.2	71.4 ± 3.3	75.8 ± 3.1	77.5 ± 3.1	68.9 ± 2.7 *
Body fat (%)	29.8 ± 1.6	28.6 ± 2	31.5 ± 1.5	31.2 ± 2	18.2 ± 1.7 *
VO_2max_ (mL^−1.^kg^−1.^min)	36.8 ± 2.1	37.3 ± 2.1	35.1 ± 1.9	34.1 ± 1.8	55.8 ± 2.7 *
HR_max_ (beats/min)	173 ± 4	178 ± 2.3	177 ± 2.2	179 ± 2.5	177 ± 3.1
RER_max_	1.16 ± 0.01	1.15 ± 0.02	1.12 ± 0.02	1.14 ± 0.02	1.09 ± 0.02

SE: standard error; *n*: number. * *p* < 0.017 versus each of the walker groups. VO_2max_ = maximal oxygen consumption rate; HR_max_ = maximal heart rate; RER_max_ = the maximal respiratory exchange ratio or VCO_2_/VO_2_.

**Table 2 nutrients-10-01718-t002:** Performance data for the 45-minute walking and 2.5-h running sessions.

Performance Variable	Walkers (*n* = 41) (45 min)	Runners (*n* = 19) (2.5 h)
VO_2avg_ (L/min)	1.65 ± 0.62	2.62 ± 1.29 *
VO_2_ (% max)	62.2 ± 0.9	69.2 ± 1.2 *
HR_avg_ (bpm)	132 ± 2.5	143 ± 4.0 *
HR (% max)	74.3 ± 1.3	80.9 ± 2.4 *
RPE_avg_	11.0 ± 0.2	11.3 ± 0.3
Weight Change (kg)	0.08 ± 0.09	−1.9 ± 0.2 *

* *p* < 0.05, walkers versus runners. RPE_avg_ = average rating of perceived exertion.

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
