# Peer review of "Increased Plasma Levels of Gut-Derived Phenolics Linked to Walking and Running Following Two Weeks of Flavonoid Supplementation"

_nutrients, 2018, doi:10.3390/nu10111718_

Reviewer 1 Report

This is a well written manuscript, applying very well validate analytical techniques and characterized by an interesting design exploring a somehow novel aspect in the framework of phenolic bioavailability.

My minor comments follow:

- The study design is clear....but I think a graphical representation might help the reader better understanding the whole concept.

- The quality of the figure is very low. And I'm not talking about dpi or definition, but of aspect. It would be nice to redo the figures in a less rough fashion. Apologies if this is no science, but it would make the whole manuscript read more pleasantly.

Author Response

REVIEWER #1

This is a well written manuscript, applying very well validate analytical techniques and characterized by an interesting design exploring a somehow novel aspect in the framework of phenolic bioavailability.

RESPONSE: Thank you for taking the time and effort to review our paper.

My minor comments follow:

- The study design is clear....but I think a graphical representation might help the reader better understanding the whole concept.

RESPONSE: A study design figure has been added to the paper.

- The quality of the figure is very low. And I'm not talking about dpi or definition, but of aspect. It would be nice to redo the figures in a less rough fashion. Apologies if this is no science, but it would make the whole manuscript read more pleasantly.

RESPONSE: Our research group met and discussed your request. We feel that in the PDF format, the colored graphs depict the results in good fashion (but perhaps not if printed in black and white). We decided to keep the graphs as they are (to keep the color theme consistent throughout).

Reviewer 2 Report

Nieman DC et all aimed to determine whether the combination of two weeks flavonoid supplementation and one acute 45 minutes brisk walking could enhance the translocation of gut-derived phenolic into circulation. To achieve their objective, they used a complex design combining a double blinded placebo controlled random sample with a control group not randomized. Here are my comments:

The study design is extremely complex, and difficult to understand as is. A figure showing the different temporal steps, including the blood samples will be very helpful for the reader. The authors should also follow the CONSORT guidelines. There is not flow chart for example. I did not really understand the utility of the control group (the runners). They removed to the study design the benefits of the randomization.

The method section should be more structured. 

The authors have measured 76 metabolites. Sometimes, they decide (on which basis?) to present the results of one metabolite, and soon after, of 15 metabolites, then 3 and 32 metabolites. It worth explain to the reader why.

The authors should consider the fact that some baseline characteristics are different across the comparison groups. This this reason, they have to account (by some way or another) of the baseline values. For example, table 2 may show the changes between post- and pre- exercise values in performance characteristics.

Was the trial registered? If yes, the authors should provide the registration number.

Table 1 compare men and women, but statistical analyses are not stratified on gender. This table is informative, but less than a table comparing general (these in table 1) and nutritional characteristics in the four groups of comparison. Even though the differences are not significant (the small sample size could be an explanation), it is important for the reader to see the differences in all the four groups. 

I do not understand the statistical analyses performed in figure 1. Also, as the participants in the runners’ group are not randomized, we cannot rollout the fact that these significant results may be due to non-controlled factors. At baseline, they have significantly different characteristics which are not accounted for, not talking about unknown characteristics.

 Author Response

REVIEWER #2

Nieman DC et all aimed to determine whether the combination of two weeks flavonoid supplementation and one acute 45 minutes brisk walking could enhance the translocation of gut-derived phenolic into circulation. To achieve their objective, they used a complex design combining a double blinded placebo controlled random sample with a control group not randomized. Here are my comments:

RESPONSE: Thank you for taking the time and effort to review our paper.

 The study design is extremely complex, and difficult to understand as is. A figure showing the different temporal steps, including the blood samples will be very helpful for the reader. The authors should also follow the CONSORT guidelines. There is not flow chart for example. I did not really understand the utility of the control group (the runners). They removed to the study design the benefits of the randomization.

RESPONSE: A study design figure has been added to add more clarity. We added more information in section 2.1 regarding the study participants that entered into the study, the number per group completing all study requirements, and the reason 4 walkers and 2 runners dropped out. We did not add a CONSORT figure because we feel this added information is sufficient (and the number of figures and tables in the paper is already high). We also added more rationale regarding the inclusion of the runner group.

 The method section should be more structured. 

RESPONSE: We reorganized and restructured the methods section in accordance with your request.

 The authors have measured 76 metabolites. Sometimes, they decide (on which basis?) to present the results of one metabolite, and soon after, of 15 metabolites, then 3 and 32 metabolites. It worth explain to the reader why.

RESPONSE: We improved this statement in the results section:

"Fifteen metabolites were identified, grouped (to improve the ability to detect intervention effects), and compared between the 4-study groups and runner comparator group."

Also note this statement:

" An additional analysis showed that this difference was primarily driven by greater runner levels for 7 plasma gut-derived phenolics (P<0.001)…"< span="">

 Many of the gut-derived phenolics are present in plasma at very low levels. Grouping of these phenolics improves the ability to detect intervention effects.

 The authors should consider the fact that some baseline characteristics are different across the comparison groups. This this reason, they have to account (by some way or another) of the baseline values. For example, table 2 may show the changes between post- and pre- exercise values in performance characteristics.

RESPONSE: Baseline characteristics were not different between the walking groups.  Notice this statement (now highlighted in red):

"A separate analysis of the four subgroups randomized among the N=77 walkers [placebo-sit (N=16), placebo-walk (N=20), flavonoid-sit (N=20), flavonoid-walk (N=21),] showed no significant differences for any of the variables listed in Table 1." 

 There were differences between the walkers and runners, but that is because we used the runners as a comparator group by design (i.e., to determine chronic exercise effects on gut-derived phenolics). Also, the walking and running bouts were very different by design (i.e., to compare responses to 45-minute brisk walking and 2.5 h intensive running).

 Was the trial registered? If yes, the authors should provide the registration number.

RESPONSE: Yes, and this has been added.

 Table 1 compare men and women, but statistical analyses are not stratified on gender. This table is informative, but less than a table comparing general (these in table 1) and nutritional characteristics in the four groups of comparison. Even though the differences are not significant (the small sample size could be an explanation), it is important for the reader to see the differences in all the four groups. 

RESPONSE: We feel the male and female comparisons in Table 1 are needed so that readers have a full awareness of our study participants. We added the nutritional data in a supplemental table (S2) in response to your request. We did not compare gut derived phenolics between genders because this study was not powered for that analysis.

 I do not understand the statistical analyses performed in figure 1. Also, as the participants in the runners’ group are not randomized, we cannot rollout the fact that these significant results may be due to non-controlled factors. At baseline, they have significantly different characteristics which are not accounted for, not talking about unknown characteristics.

RESPONSE:  The statistical section describes the analysis for the hippurate figure:

"The plasma metabolite data (both single metabolites and grouped metabolites) were analyzed using the generalized linear model (GLM), and a 5 (groups) x 4 (time) repeated-measures ANOVA, between-participants design (IBM SPSS Statistics for Windows, Version 24.0, IBM Corp, Armonk, NY, USA)."

 The runner group was a comparator group that was selected to run in parallel with the walker groups (as more fully described in the methods section). Thus the statistical model compares 5 distinct groups over time.

 Round  2

Reviewer 2 Report

I am not satisfied with the response of the authors. 

 Particularly, the figure added on the study design (Figure 1) do not show that there were five groups at the end of the study. There was no structuration at all of the methods section as said by the authors. I asked for a table showing the characteristics of participants (those in table 1 and nutritional characteristics) in the four groups even if they were not significantly different (probably because of the low number of participants). In supplementary table 2 (Excel sheet), I computed the mean energy intake of walkers and runners, I did not obtain the same values as in highlighted in green on the Excel sheet.

Author Response

Particularly, the figure added on the study design (Figure 1) do not show that there were five groups at the end of the study.

RESPONSE: Figure has been changed in accordance with your recommendation.

 There was no structuration at all of the methods section as said by the authors.

RESPONSE: We extensively restructured the methods section in the first revision, and have added more structure to the Targeted Metabolomics Analysis section.

 I asked for a table showing the characteristics of participants (those in table 1 and nutritional characteristics) in the four groups even if they were not significantly different (probably because of the low number of participants).

RESPONSE: We changed Table 1 in accordance with your recommendation. We also formed a summary table of nutrient intake data for each of the five groups, and included this in Table S2.  We felt that this table would take up too much room in the paper and detract from the primary outcome measure (gut derived phenolics).

In supplementary table 2 (Excel sheet), I computed the mean energy intake of walkers and runners, I did not obtain the same values as in highlighted in green on the Excel sheet.

RESPONSE: We fixed the formula for the group comparisons.  Thank you for finding this error.

Round  3

Reviewer 2 Report

The authors have fulfilled my requests. 

For table 1, as well as for supplementary table 2, A global test has to be performed first (alternative hypothesis: at last one measure is different from others) before the two by two comparisons (to limit type 1 error). For the supplementary table 2, see the document attached (only the p values highlighted in green).

Author Response

The authors have fulfilled my requests. 

For table 1, as well as for supplementary table 2, A global test has to be performed first (alternative hypothesis: at last one measure is different from others) before the two by two comparisons (to limit type 1 error). For the supplementary table 2, see the document attached (only the p values highlighted in green).

RESPONSE:  In response to your recommendation, we conducted comparisons between the walker groups and runners (subject characteristics, performance data, nutrient intake, and other selected variables) using one-way ANOVA, with post-hoc tests using Tukey's HSD and statistical differences accepted when the P-value was ≤0.05. We added statements to the results section as follows:

No significant differences were found for any of the variables listed in Table 1 among the five groups except for significantly higher maximal oxygen consumption rates (VO2max) and lower body weight and body fat percentages for the runner group compared to each of the walker groups…. 

Three-day food records collected before the study began revealed no significant group differences using oneway ANOVA in energy, macronutrient, micronutrient intake, and total flavonoid intake except for cholesterol (F=3.05, p=0.021, with Tukey's HSD post-hoc tests showing no significant group differences) (supplemental Table S2).

NOTE that only cholesterol had a significant oneway ANOVA test result, but Tukey's HSD revealed no group differences.